# Green and Cost-Effective Separation of Cadmium from Base Metals in Chloride Medium with Halide-Loaded Anion Exchanger

Yanlin Zhang [1,2], Xiaofei Duan [2], Chaoquan Hu [1,3], Guanshang Du [1] and Yong Wang [3,4,5,*]

1 Zhongke Nanjing Institute of Green Manufacturing Industry, Nanjing 211135, China
2 School of Chemistry, The University of Melbourne, Parkville, Melbourne, VIC 3010, Australia
3 Institute of Process Engineering, Chinese Academy of Sciences, Beijing 100190, China
4 National Engineering Research Centre of Green Recycling for Strategic Metal Resources, Beijing 100190, China
5 The University of Chinese Academy of Sciences, Beijing 101400, China
* Correspondence: wangyong@ipe.ac.cn

**Abstract:** A novel strategy for the separation of cadmium from base metals in chloride medium by anion exchange based on the formation of cadmium halo-complexes is presented in this article. Instead of using excess $Br^-$ or $I^-$ in the feed solution, the anion exchange resin or fibre was first preloaded with $Br^-$ or $I^-$ for $Cd^{2+}$ adsorption from $Cl^-$ solution. Thus, the consumption of iodide or bromide was minimized while promising high adsorption stability of $Cd^{2+}$. The adsorption thermodynamics and kinetics of $Cd^{2+}$ were studied. The adsorption thermodynamics results were processed with Langmuir and Freundlich models and adsorption kinetics results were treated with Pseudo first order model, Pseudo second order model and intraparticle diffusion model. The adsorption mechanism was studied with the aid of FTIR and elemental analysis. Three strategies were applied for the stripping of $Cd^{2+}$, including the use of water for disequilibrating its halo-complex formation, the formation of positively charged species with ammonia, and the formation of positively charged chelate with multidentate ligand ethylenediamine. On the bases of these results, separation performance was investigated with strongly basic anion exchange resin and fibre, and it was shown that $Cd^{2+}$ can be efficiently separated from base metals in a wide range of acidity, and that pure $Cd^{2+}$ solution can be obtained, thus providing a robust, environmentally friendly, and economic technology for its separation.

**Keywords:** cadmium separation; base metals; halide-loaded anion exchanger; chloride medium

## 1. Introduction

Cadmium commonly exists in natural minerals. It is a valuable resource, but also an unwanted impurity in industrial products, especially in some metals, and hence, its removal and/or recovery are common issues in the production of these products [1–3]. Typical examples with industrial significance include the removal of cadmium from phosphoric acid used in fertilizer production [4], cadmium removal from effluents and process streams for environmental protection [5], and recycling of spent batteries for recovery of the valuable metals [6–10]. One of the largest areas for the separation of cadmium, however, is in the production of some metals from their ores, particularly lead and zinc, where cadmium commonly occurs with these metals and must be removed during the refining processes to obtain pure metal products [2,11]. Separation of Cd and other co-existed elements from their ores often generates large amounts of solid residues containing different metal elements such as Cd, Cu, Pb, Ni, etc. [12,13]. Great efforts have been made to make use of these secondary resources, and cost-effective extraction techniques for these elements have been a hot topic in metallurgical study [11,14].

Although various techniques have been studied for the separation and/or recovery of Cd from mineral resources and much progress has been made in industrial application, many of the techniques still have some shortcomings in terms of complexity, efficiency, facility investment, operational parameters, cost-effectiveness, and pollution. Pyrometallurgical processes were conventionally used for the separation and recovery of Cd from various resources and are still applied in many industries. For instance, in the recycling of spent Ni-Cd battery, 99.92% Cd is separated from the matrix at 900 °C with coal (anthracite) as a carbonaceous material, whereas Ni and Co form an alloy [6]. Distillation has been used for the recovery of Cd under high temperature with three processes, i.e., SNAM—SAVAM, SAB—NIFE, and INMETCO [6,15]. The use of vacuum can facilitate metallurgical separation with high efficiency and low pollution [16]. Although the pyrometallurgical techniques have the advantages of high reaction rate, high overall efficiency and productivity and need fewer chemicals than hydrometallurgical techniques, the co-existed metals with high boiling point usually form new slag after Cd separation and the recovery of valuable metals from the slag still needs hydrometallurgical refining [6]. In addition, such pyrometallurgical processes often need waste gas treatment because harmful gas is released, and such techniques are not profitable for low-grade resources considering the high energy consumption and low valuable metal productivity [6].

Compared with pyrometallurgical processes, hydrometallurgical technologies are more appreciated from economic and environmental perspectives in the separation of cadmium from many resources because less energy is consumed and less dust collecting and gas cleaning are needed [3]. Hydrometallurgical technologies developed for this purpose include precipitation [17,18], cementation [14,19], electrodeposition [8,20], solvent extraction [2,11,21–27], membrane technology [28–31], imprinted sorbent separation [32–34], water-based two-phase system extraction [35], selective leaching [36,37], and solid phase extraction [38,39]. Some of these techniques, such as imprinted sorbent separation, have only been practiced in lab-scale research due to their technical limitations. In comparison, several technologies including cementation, solvent extraction, and solid phase extraction have found wide industrial application. Cementation is often employed in industry to remove Cd from Zn concentrate [14]. However, recovery of metal values including Cd from secondary resources such as Cd-bearing purification filter cake requires a series of dissolution and re-cementation stages, usually culminating in the electrowinning of Cd metal to obtain a product of sufficient purity for commercial demands [40]. In addition, this process is not applicable to Cd removal from moderately acidic medium (pH 1.5–2) because unacceptable quantities of zinc dust would be consumed for neutralizing the excess acid [41].

Solvent extraction has been extensively studied and applied to the separation of Cd from base metals [42]. The separation efficiency is basically determined by the properties of the coexisted elements and the performance of the extractants. In particular, due to their close physical and chemical properties, separation of Cd from Zn is highly challenging [43], especially when a single extractant is used [2,11,23,25], despite the fact that high separation efficiency has been achieved by careful optimization in some examples [3,23]. As a consequence, synergistic extraction has received more attention in the recent decades and better selectivity has been achieved [2,21–23]. For example, Preston and co-workers [21] found that the addition of 0.50 M triisobutylphosphine sulphide (TIBPS or Cyanex 471X) to 0.50 M 3,5-diisopropylsalicylic acid (DIPSA) resulted in a strong synergistic extraction effect for $Cd^{2+}$, as indicated by a shift of $pH_{50}$ (the pH at which 50% of metal is extracted) by 1.9 pH units, while $pH_{50}$ shifts of less than 0.6 pH unit for $Zn^{2+}$, $Pb^{2+}$, $Mn^{2+}$, $Fe^{3+}$, $Co^{2+}$, $Ni^{2+}$, $Ca^{2+}$, and $Mg^{2+}$ were observed, which thus enabled the selective extraction of $Cd^{2+}$ from these metals. In a study by Moradkhani and co-workers [2], $pH_{50}$ and $\Delta pH_{50}$ values were determined and compared for a number of organic systems for synergistic extraction of $Cd^{2+}$ from $Zn^{2+}$. The largest $\Delta pH_{50(Zn–Cd)}$ value was 1.53 pH units obtained with nonylsalicylic acid (HRJ-4277)/TIBPS system, implying excellent separation of $Cd^{2+}$ from $Zn^{2+}$ with this system. However, the $pH_{50}$ values for $Cd^{2+}$ were still not low enough for acidic

industrial materials and hence, neutralization is often necessary before the extraction is carried out [2,21–23]. Moreover, the $\Delta pH_{50(Cu-Cd)}$ and $\Delta pH_{50(Pb-Cd)}$ values are small [11,21] and multi-stage extraction is often needed to realize satisfactory Cd separation [11]. In addition, many of the extractants, especially organophosphorus and organosulfur compounds, are highly toxic.

Solid phase extraction (SPE), especially ion exchange, has exhibited many advantages over other technologies, including simpler operational processes without the use of volatile and toxic solvents, lower pollution than solvent involved extraction processes, avoidance of sludge generation, milder operating conditions, and tolerance for wider pH range [44]. A number of studies on both anion and cation exchange methods based on halo-complex formation have been reported for the separation of Cd from other metals [1,38,39,41,45]. In anion exchange methods, $Cd^{2+}$ was usually adsorbed by strong base anion exchange resin in the form of $[CdX_4]^{2-}$ in the presence of excess of $X^-$ and subsequently stripped with highly concentrated $NaNO_3$ or $HNO_3$ [38,45]. In cation exchange methods, $Cd^{2+}$ and other metal cations were adsorbed onto a strongly acidic cation exchanger and $Cd^{2+}$ was selectively eluted with halide solution [46–48]. Alternatively, $Cd^{2+}$ passes through the cation exchange resin in the form of $[CdI_4]^{2-}$ while other cations are retained by the resin due to their different stability of iodo-complexes [49]. In the anion exchange procedures, moderate to high concentrations of ligand ($X^-$) were used in the feed solutions and this often means high operating costs when using bromide or iodide, and the striping of $Cd^{2+}$ by $NaNO_3$ or $HNO_3$ is usually very slow because of the strong retention of $[CdX_4]^{2-}$ [45,50,51]. In addition, highly concentrated $HNO_3$ as the stripping agent can oxide $I^-$ adsorbed on the resin [50]. Halide solution with high concentration has also been used for the elution of $Cd^{2+}$, but the stripping was not fast and efficient [38].

Considering the above-mentioned shortcomings of the conventional techniques, it is of significance to develop a more efficient, environmentally benign and economic technique for the separation and recovery of Cd for industrial application. Therefore, a new strategy for ion exchange separation of $Cd^{2+}$ is presented in this article. The stability of $Cd^{2+}$ halo-complexes increases in the order of $[CdCl_4]^{2-} < [CdBr_4]^{2-} < [CdI_4]^{2-}$ [52]. As such, the favorable order of using halide as the ligand for $Cd^{2+}$ complexation is in the order of $I^- > Br^- > Cl^-$. In this new method, anion exchange resin is first loaded with $Br^-$ or $I^-$, and such modified resins are used for the adsorption of $Cd^{2+}$ from $Cl^-$ solution, thus consumption of bromide and iodide is minimized while achieving high adsorption performance and selectivity because of the higher affinity of $Br^-$ and $I^-$ to $Cd^{2+}$ than $Cl^-$. Such a method has been applied to lab-scale in-line separation of $Cd^{2+}$ from base metals in the digest of solid environment samples for Cd determination in conjunction with atomic absorption spectrometry with a detection limit of 3 ng $L^{-1}$ and precision of around 5% achieved [53]. However, some fundamental issues relating to the adsorption, desorption and separation processes, including thermodynamics, kinetics, and mechanism, are to be further clarified. The present work aims at elucidating these issues with experimental data. It will investigate the performance of halide preloaded strong base anion exchangers in the separation of Cd from other base metals including Mn, Fe, Co, Ni, Cu, Zn, and Pb. The adsorption isotherms and kinetics of $Cd^{2+}$ on $Cl^-$-, $Br^-$-, or $I^-$-loaded anion exchange resin and the elution strength of water, ammonia, and ethylenediamine solutions from the loaded resin and fiber will be examined. The separation performance of Cd from the above-mentioned base metals will be presented. An Amberlite IRA 900 macroporous strongly basic anion exchange resin will be tested as the adsorbent for the thermodynamics and kinetics study. At the same time, a ZB-2 strongly basic anion exchange fiber will be used for comparison in desorption and separation performance experiments. This research aims at achieving a scalable technique for industrial application.

## 2. Materials and Methods

### 2.1. Reagents and Apparatus

All chemicals are reagent grade unless otherwise stated. Amberlite IRA 900 strongly basic anion exchange resin (Sigma-Aldrich Chemie GmbH, Buchs, France) and ZB-2 strongly basic anion exchange fibre (Guilin Zhenghan Science and Technology Developing Co. Ltd., Guilin, China) were used for the study. The Amberlite IRA 900 resin has particle size of 297–841 micron, exchange capacity of 1.0 meq g$^{-1}$, and pore size of 100 nm. The ZB-2 fibre has linear density of 1.7–2.3 dtex and exchange capacity of $\geq$3 meq g$^{-1}$.

Individual $Cd^{2+}$, $Ni^{2+}$, and $Pb^{2+}$ stock solutions of 1% (*m/v*) and $Mn^{2+}$, $Fe^{3+}$, $Co^{2+}$, $Cu^{2+}$ and $Zn^{2+}$ stock solutions of 5% (*m/v*) were prepared by first dissolving their chloride or oxide in 2% hydrochloric acid. The final HCl concentration in the solutions was 0.15 M. Standard solutions of $Cd^{2+}$, $Zn^{2+}$, $Mn^{2+}$, $Fe^{3+}$, $Co^{2+}$, $Ni^{2+}$, $Cu^{2+}$ and $Pb^{2+}$ of 1000 mg L$^{-1}$ in 1 M HNO$_3$ (National non-ferrous metal and electronic materials analytical and testing center, Beijing, China) were used for calibration in spectroscopic analysis of the metals after adequate dilution. Potassium bromide and potassium iodide were purchased from Shanghai Macklin Biochemical Co., Ltd. (Shanghai, China). Ammonia solution (25–28%, Shanghai Lingfeng Chemical Reagent Co., Ltd., Shanghai, China) and ethylenediamine (Nanjing Chemical Reagent Co., Ltd., Nanjing, China) were used for $Cd^{2+}$ desorption after adequate dilution with water. Milli-Q water was used in all experiments.

An Agilent 7900 inductively coupled plasma mass spectrometer (ICP-MS) (Agilent Technologies, Inc., Tokyo, Japan) was used for the detection of the metal elements. In the separation performance experiments, a multi-channel peristaltic pump (Model ISM944, ISMATEC SA, Glattbrugg, Switzerland) was used for the percolation of sample solution, washing agent and eluent with Tygon tubing of 1.6 mm inner diameter (id). A Nicolet Summit FTIR spectrometer (Thermo Fisher Scientific Inc., Waltham, MA, USA) was used for IR spectrum measurement.

### 2.2. Preparation of Halide-Loaded Anion Exchangers

To prepare $Br^-$- or $I^-$-loaded anion exchange resin or fibre, 5 g of the exchangers were placed in 100 mL of 0.5 M KBr or KI solution for two hours under agitation with magnetic stirrer followed by filtration and thorough washing with water until no $Br^-$ or $I^-$ was detectable by reaction with AgNO$_3$. $Cl^-$ preloading was not necessary because all the commercial exchanger products were supplied in Cl-form. Once immobilized on the resin or fibre, $Br^-$ and $I^-$ were quite stable, and considerable replacement by $Cl^-$ was not observed under the experimental conditions in this study due to their higher affinity to the quaternary ammonium groups than $Cl^-$.

### 2.3. Adsorption Isotherm Study

For the preparation of adsorption isotherms, different volumes of $Cd^{2+}$ stock solution of 4.39 mg Cd mL$^{-1}$ in 0.15 M HCl were added into seven 150 mL conical glass flasks containing 0.5 g of $X^-$-loaded Amberlite IRA 900 resin each and made up to 50 mL with 0.15 M HCl solution and agitated on a HY-2A platform mixer (Guohua Changzhou Instruments Pty. Ltd., Changzhou, China) at 150 rpm for 2 h at 25 $\pm$ 2 °C. For $Cl^-$-loaded resin (Cl-resin) and $Br^-$-loaded resin (Br-resin), the quantities of $Cd^{2+}$ added to the seven flasks were 2.2, 4.4, 8.8, 17.6, 26.3, 35.1 and 43.9 mg, respectively. For $I^-$-loaded resin (I-resin), the quantities of $Cd^{2+}$ added to the seven flasks were 4.4, 8.8, 17.6, 26.3, 35.1, 43.9 and 65.9 mg, respectively. $Cd^{2+}$ in the equilibrium solutions was determined with ICP-MS after adequate dilution and calculated through standard calibration and the quantity of $Cd^{2+}$ adsorbed by the resin was calculated by mass balance.

### 2.4. Adsorption Kinetics Study

All the kinetic experiments were conducted using 100 mL of $Cd^{2+}$ solution in 0.15 M HCl. Duplicate tests were conducted for each experiment and $Cd^{2+}$ concentrations for the two parallel experiments were 40 mg L$^{-1}$ and 80 mg L$^{-1}$, respectively. The solutions were

added to 150-mL Pyrex glass conical flasks. 0.5 g (dry) of $X^-$-loaded Amberlite IRA 900 resin was added into each flask and the mixture was agitated with a magnetic stirrer at 600 rpm. Aliquots of 200 μL of the 40 mg $L^{-1}$ or 100 μL of the 80 mg $L^{-1}$ solution were withdrawn at certain time intervals for $Cd^{2+}$ determination by ICP-MS after dilution with water and the mass of adsorbed Cd versus time relationship was constructed.

### 2.5. Effect of Acidity on $Cd^{2+}$ Adsorption

To investigate the effect of acidity on the adsorption efficiency of $Cd^{2+}$, 0.5 g of dry X-preloaded anion exchange resin was packed into a plastic syringe of 9 mm in diameter, producing 1 mL volume after saturation with water. 20 mL feed solution of 0.1% $Cd^{2+}$ was loaded into the column and drained dropwise at a flow rate of 0.5 mL $min^{-1}$. Feed solutions with HCl concentrations of 0.1 M, 0.5 M, 1.0 M, 3.0 M, and 6.0 M were compared. The $Cd^{2+}$ retention efficiency at different acidities was measured by analyzing the residual $Cd^{2+}$ concentration in the effluents by ICP-MS.

### 2.6. Stripping Agent

Different stripping agents were compared with online method using an ion exchange column and a peristaltic pump. 0.5 g dry $X^-$-loaded Amberlite IRA 900 resin or ZB-2 fibre was packed into a plastic syringe of 9 mm in diameter, producing 1 mL volume after saturation with water. The column was loaded with $Cd^{2+}$ by percolating 20 mL solution containing 0.2% $Cd^{2+}$ in 0.15 M HCl through it at 1 mL $min^{-1}$, followed by washing with 20 mL 0.10 M HCl to remove the un-adsorbed $Cd^{2+}$. Adsorbed $Cd^{2+}$ was stripped with different agents including water, 0.5 M ammonia and 0.3 M ethylenediamine (En) solutions at 1 mL $min^{-1}$. The eluate was collected in fractions of 2 mL (water and $NH_3$) or 1 mL (En). $Cd^{2+}$ in the effluents and eluates was determined by ICP-MS and the retention efficiency of $Cd^{2+}$ can be calculated by mass balance according to the $Cd^{2+}$ concentration in the effluents, while the desorption profile can be established with the $Cd^{2+}$ concentrations in the eluate fractions.

### 2.7. Separation Performance

Separation performance was investigated in the same way as in the stripping agent comparison with a synthetic feed solution containing 0.1% $Cd^{2+}$, 1% $Zn^{2+}$, 1% $Cu^{2+}$, 0.1% $Pb^{2+}$, 0.1% $Mn^{2+}$, 0.5% $Fe^{3+}$, 0.1% $Co^{2+}$, and 0.1% $Ni^{2+}$ (*w/v*) in 0.15 M HCl. Cd and other elements in the effluents and eluates were determined with ICP-MS so that the retention efficiency and recovery of each element and the separation efficiency of Cd from each co-existed element can also be calculated. The above concentration values were chosen just for fundamental study, rather than for any special raw material from industry. Cd usually exists in mineral resources as a minor component. As such, it was added into the synthetic feed solution with a lower concentration than some major elements such as Zn, Cu and Fe in minerals.

## 3. Results and Discussion

### 3.1. Selection of Ligand and Concentration in the Feed Solution

The presence of $X^-$ in the feed solution is important for the adsorption of $Cd^{2+}$ on an anion exchanger due to its complexation with $Cd^{2+}$ forming the negatively charged species $[CdX_4]^{2-}$. In hydrometallurgical processing of minerals, base metals are often leached with HCl or $H_2SO_4$ [3,8–10,18,22,54]. After leaching with HCl, $Cd^{2+}$ and $Cl^-$ ions can form stable $[CdCl_4]^{2-}$ for anion exchange separation. When leached with $H_2SO_4$, halide ions ($Cl^-$, $Br^-$ or $I^-$) is to be added as ligand to form complex with $Cd^{2+}$ for its adsorption by anion exchanger. Although $Br^-$ and $I^-$ form more stable complexes with $Cd^{2+}$ which is beneficial for the adsorption of $Cd^{2+}$ by anion exchanger [45,50], this advantage is compromised by the high cost of bromide and iodide salts or acids. Therefore, HCl was chosen as the sample medium while $Br^-$ or $I^-$ was pre-loaded onto the anion exchangers in this study, thus improving the adsorption performance of commercial $Cl^-$-formed anion exchanger

while minimizing consumption of $Br^-$ or $I^-$. In the adsorption of $Cd^{2+}$ onto the $X^-$-loaded exchanger, the exchanger may offer two X atoms for coordinating $Cd^{2+}$ while the feed solution provides two $Cl^-$ ions for the formation of $[CdX_4]$ as detailed below. Therefore, the concentration of $Cl^-$ in the feed solution should be a determinant for the adsorption. This has been verified by our experiment which showed that the retention efficiency and adsorption rate of $Cd^{2+}$ on the $X^-$ loaded resins increased as increasing HCl concentration. The retention efficiency culminated with 0.15 M HCl and levelled off with higher HCl concentration. Based on these facts, the feed solution was prepared in a medium containing 0.15 M HCl in the subsequent experiments.

### 3.2. Adsorption Isotherms and Mechanism

Several models can be used to describe the variations of the adsorption data. The two most frequently used ones for dilute solutions are the Langmuir and Freundlich isotherms [55]. Langmuir treatment is based on three assumptions that: (a) maximum adsorption corresponds to a saturated monolayer of solute molecules or ions on the adsorbent surface, (b) the energy of adsorption is constant, and (c) there is no transmigration of adsorbate in the plane of the surface [56], which is represented by:

$$q_e = K_a q_m C_e / (1 + K_a C_e) \tag{1}$$

or, in the linear form of

$$C_e/q_e = 1/q_m K_a + C_e/q_m \tag{2}$$

where $q_e$ is the amount of adsorbate adsorbed per unit mass of adsorbent (mg/g) at equilibrium state and $C_e$ is the liquid phase concentration of the adsorbate (mg/L) at equilibrium. $q_m$ and $K_a$ are Langmuir constants representing adsorption capacity and equilibrium constant, respectively.

The Freundlich equation is derived to model multilayer adsorption on heterogeneous surfaces, which is expressed as [56]:

$$q_e = K_F C_e^{1/n} \tag{3}$$

or, in the linear form of

$$\log q_e = \log K_F + (1/n)\log C_e \tag{4}$$

where $q_e$ and $C_e$ have the same meaning as in the Langmuir model, while $K_F$ and $n$ are Freundlich constants related to adsorption capacity and adsorption intensity, respectively.

The equilibrium data of $Cd^{2+}$ adsorption on $X^-$-loaded Amberlite IRA 900 resin and the Langmuir and Freundlich isotherms are illustrated in Figure 1, and the isotherm parameters obtained from linear analysis of $Cd^{2+}$ adsorption by the two models are listed in Table 1.

The $K_a$ and $K_F$ data in Table 1 show that the adsorption performance of $Cd^{2+}$ on the halide-loaded resins rose in the order of Cl-resin < Br-resin << I-resin. This is in accordance with the stability of $[CdX_4]^{2-}$ complexes [52]. The $K_a$ and $n$ values show that all the adsorptions are favorable [57].

The graphs in Figure 1 show that the adsorption of $Cd^{2+}$ on Cl-resin and Br-resin can be described by Langmuir model while the data for I-resin can be described by Freundlich model. This may indicate that different adsorption mechanisms are followed. On the Cl-resin and Br-resin, monolayer adsorption of $Cd^{2+}$ halo-complex should happened which can be expressed as $[Cd(Cl_S)_2(Cl_L)_2]$ and $[CdBr_2(Cl_L)_2]$, where $Cl_S$ and $Cl_L$ refer to Cl provided by the preloaded Cl on the resin and the liquid phase, respectively. In comparison, multilayer adsorption may have occurred on I-resin. Another possibility in I-resin adsorption is the formation of mixed-ligand complexes of different compositions on the resin surface, such as $[CdI(Cl)_3]$ and $[CdI_2(Cl)_2]$ as discussed below. The former one may augment the adsorption capacity of the I-resin. The adsorption capacity data ($q_m$) in Table 1 also indicate the different adsorption performance of the three resins with the I-resin having

significantly higher capacity. It is worth mentioning that the difference should be more profound by taking the use of the same mass of halide modified resins in the experiment and the different atomic weights of Cl, Br, and I into account.

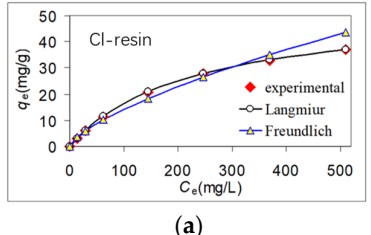
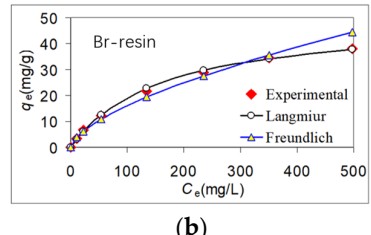
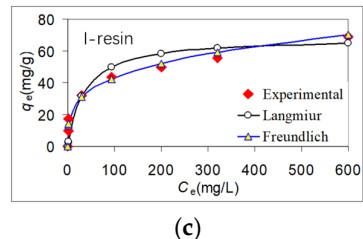

(**a**)    (**b**)    (**c**)

**Figure 1.** Theoretical and experimental non-linear isotherm plots of $Cd^{2+}$ adsorption on halide-loaded Amberlite IRA 900 resins. (**a**) Cl-resin; (**b**) Br-resin; (**c**) I-resin.

**Table 1.** Isotherm parameters obtained from linear analysis of $Cd^{2+}$ adsorption on $X^-$-loaded resins.

|  |  | $K_a$ | $q_m$(mg/g) | $R^2$ |
|---|---|---|---|---|
| Langmuir model | Cl-resin | 0.0045 | 53.19 | 0.9991 |
|  | Br-resin | 0.0063 | 54.75 | 0.9972 |
|  | I-resin | 0.0277 | 68.97 | 0.9777 |
|  |  | $K_F$ | $n$ | $R^2$ |
| Freundlich model | Cl-resin | 0.5867 | 1.446 | 0.9842 |
|  | Br-resin | 0.8734 | 1.581 | 0.9835 |
|  | I-resin | 12.325 | 3.671 | 0.9882 |

It has been established that $Cd^{2+}$ complexes with mixed halide ligands are formed with high stability in multiple-halide aqueous solution [58,59]. For instance, in the presence of $Cl^-$ and $I^-$, $Cd^{2+}$ could form different species such as $[CdClI_3]^{2-}$, $[CdCl_2I_2]^{2-}$ and $[CdCl_3I]^{2-}$, depending on the relative concentrations of $Cl^-$ and $I^-$. As such, we can assume that in the ion exchange process in this study, the simultaneous presence of different halide anions could also result in the formation of different $[CdX_4]^{2-}$ species with mixed halide ligands. To validate these assumptions, anion exchange resin with different forms (C-N(CH$_3$)$_3$-OH, C-N(CH$_3$)$_3$-Cl, C-N(CH$_3$)$_3$-I), and Cd-loaded X-resins were characterized with FTIR and the results are presented in Figure S1. When the $Cl^-$ ions on the resin were replaced with $OH^-$, an intense peak at 1610 cm$^{-1}$ was observed in the IR spectrum (Figure S1b). It is due to the deformation vibration of C-N(CH$_3$)$_3$ and O-H bonds [60]. In -Cl form, the peak is much weaker with a shoulder peak at about 1618 cm$^{-1}$ (Figure S1a). Similarly, when the C-N(CH$_3$)$_3$ was associated with $I^-$, the peak was also much weaker than that with -OH groups but without the shoulder peak and the peak is more intense than that of the Cl-resin (Figure S1c). When the resin was loaded with $[CdCl_4]^{2-}$, the peak was also weaker than the -OH formed resin with the shoulder peak observed (Figure S1d), and when the resin was loaded with $[CdI_4]^{2-}$ by adsorbing $Cd^{2-}$ from 0.1 M KI solution with I-resin, the shoulder peak disappeared but the intensity is higher than that of the $[CdCl_4]$ loaded resin (Figure S1g). In comparison, when the resin was loaded with Cd complex with mixed halide ligands ($Cl^-$ and $I^-$), the peak profile and relative intensity are something between those of $[CdCl_4]^{2-}$ and $[CdI_4]^{2-}$ loaded resins (Figure S1e,f)). These results could be interpreted by the formation of mixed-ligand complexes of $Cd^{2+}$ under different conditions. By using different $Cl^-$ concentrations in the feed solution for $Cd^{2+}$ adsorption by the I-resin, different species of Cd complexes, such as $[CdCl_2I_2]^{2-}$ and $[CdCl_3I]^{2-}$ could be formed. This is in line with the situation in solution chemistry [61]. It should be noted that $Cd^{2+}$ ions were loaded from highly acidic solutions onto the resins and hence, the residual $OH^-$ groups should be negligible. As such, the peak at 1610 cm$^{-1}$ after loading with $[CdX_4]^{2-}$ should not be considered as a result of the presence of $OH^-$ group associated with the quaternary ammonium groups, instead, it could reflect the change of

the deformation vibration of C-N(CH$_3$)$_3$ group. The spectroscopic results should therefore indicate the effect of the [CdX$_4$]$^{2-}$ species on the C-N bond.

The change of the intense peak at 3374 cm$^{-1}$, corresponding to the stretching vibration of the O-H bond of the -OH group in the resin (Figure S1b), after loading the resin with different halide or Cd$^{2+}$ halo-complex species, also indicates the influence of the Cd$^{2+}$ halo-complex species. In particular, by loading with [CdCl$_4$]$^{2-}$ or [CdI$_4$]$^{2-}$, the peak shifted to 3390 cm$^{-1}$ and 3500 cm$^{-1}$, respectively. By loading with [CdCl$_x$I$_y$]$^{2-}$ with increased $x$ value and decreased $y$ value (adsorbed from 0.1 M and 1 M HCl, respectively), the peak shifted towards lower wavenumber from 3572 cm$^{-1}$ to 3500 cm$^{-1}$. This trend of wavenumber shifting is in accordance with the possible change in the complex composition.

However, the above spectroscopic results are not sufficient as a justification for the assumed complex species. For this reason, elemental analysis was performed by ICP-MS. Br-resin and I-resin were first saturated with Cd$^{2+}$ by adsorption from HCl solution. After removing the excess Cd$^{2+}$ in the resin micropores and voids with 0.1 M HCl, the resins were washed with water followed by complete stripping of Cd$^{2+}$ with 0.3 M ethylenediamine (En). Br$^-$ and I$^-$ in the resins were completely stripped with 0.1 M KOH solution. Br, I and Cd in the respective water, En eluates and KOH solutions were determined by ICP-MS and the molar ratios of Br/Cd and I/Cd were calculated. The results showed that the Br/Cd ratio was 2.04 ± 0.03, and the I/Cd ratio was 1.89 ± 0.05, indicating that [CdBr$_2$Cl$_2$]$^{2-}$ and [CdI$_2$Cl$_2$]$^{2-}$ should be the dominant species on the Br-resin and I-resin, respectively. The I/Cd ratio lower than 2 may imply that species corresponding to [CdICl$_3$]$^{2-}$ exists on the resin, no matter if monolayer or multi-layer adsorption took place.

### 3.3. Adsorption Kinetics

#### 3.3.1. Effect of Contact Time

Figure 2 shows the time courses of Cd$^{2+}$ adsorption on X$^-$-loaded Amberlite IRA 900 resins by using 40 mg L$^{-1}$ and 80 mg L$^{-1}$ as the initial Cd$^{2+}$ concentrations. Adsorption processes could follow different mechanisms and be controlled by various factors, such as film diffusion, intra-particle diffusion or chemical reaction. In order to elucidate the adsorption process, several adsorption models are applicable for evaluation of the experimental data. In this study, Lagergren's pseudo-first order kinetic model, the pseudo-second order kinetic model, and intra-particle diffusion model were applied for the data treatment.

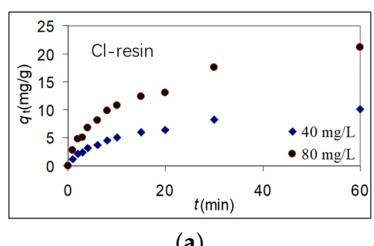
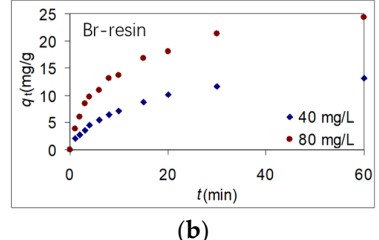
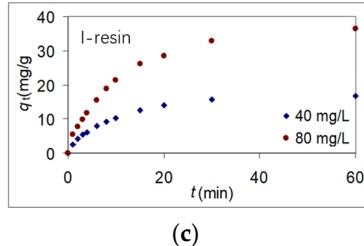

(a)        (b)        (c)

**Figure 2.** Mass-time profiles of Cd$^{2+}$ adsorption on X$^-$-loaded resins. (**a**) Cl-resin, (**b**) Br-resin, (**c**) I-resin.

#### 3.3.2. Pseudo-First Order Kinetics

The linearized form of the pseudo-first order rate equation by Lagergren [55] is given as

$$\ln(q_e - q_t) = \ln q_e - k_1 \times t \tag{5}$$

where $q_e$ and $q_t$ are the amounts of the metal ion adsorbed (mg g$^{-1}$) at equilibrium and at time $t$ (min), respectively and $k_1$ is the adsorption rate constant (min$^{-1}$). The linear regression of $\ln(q_e - q_t)$ versus $t$ gives a straight line and the rate constant ($k_1$) can be calculated from its slope. The Lagergren's plots are given in Figure 3. The $k_1$, $q_e$ and $R^2$ obtained from this model are listed in Table 2. It shows that the Lagergren's pseudo-first order model fits the experimental data well and the adsorption rate constants increase in

the order of Cl-resin < Br-resin < I-resin. The theoretical and experimental $q_e$ values are also in good agreement, implying the adsorption process can be assumed as a first order reaction. Obviously, the concentration of $[Cd(Cl)_4]^{2-}$ at the surface of the resin is the major controlling factor of the adsorption rate. The data with different initial $Cd^{2+}$ concentrations also support the fitting of the pseudo-first order model. However, the difference in the rate constant with different resins implies that the adsorption process is not a pure first order reaction. Ligand exchange could also play a significant role, i.e., ligand exchange rate between $X_S$ and $Cl_L$ increases in the order of Cl < Br < I. $X_S$ and $Cl_L$ denote the pre-immobilized $X^-$ on the resin and $Cl^-$ ions in the liquid phase, respectively.

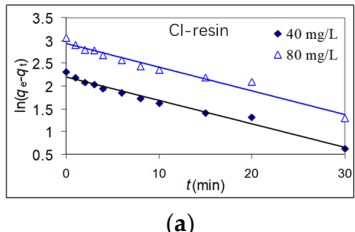 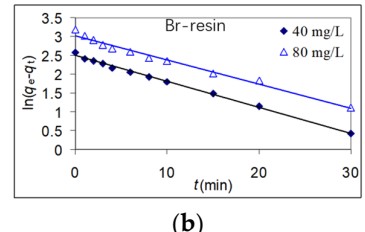 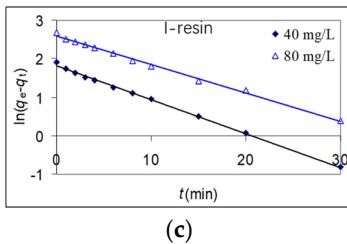

(**a**)　　　　　　　　　　　　　　　(**b**)　　　　　　　　　　　　　　　(**c**)

**Figure 3.** Pseudo-first order plots of $Cd^{2+}$ adsorption on $X^-$-loaded resins. (**a**) Cl-resin, (**b**) Br-resin, (**c**) I-resin.

**Table 2.** Kinetic parameters with pseudo-first order, pseudo-second order and intra-particle diffusion model analysis.

| Resin | $C_0$ (mg/L) | Pseudo-First Order Parameters | | | | Pseudo-Second Order Parameters | | | Intra-Particle Diffusion Rate Constants | |
|---|---|---|---|---|---|---|---|---|---|---|
| | | $q_e$ (exp., mg/g) | $q_e$ (Theor. mg/g) | $k_1$ (min$^{-1}$) | $R^2$ | $k_2 \times 10^3$ (g/mg$^{-1}$ min$^{-1}$) | $q_e$ (Theor., mg/g) | $R^2$ | $k_{id}$ (µg g$^{-1}$ min$^{-1/2}$) | $R^2$ |
| Cl-resin | 40 | 10.1 | 8.98 | 0.0513 | 0.9792 | 7.3 | 11.7 | 0.9808 | 1.36 | 0.9826 |
| | 80 | 21.2 | 18.8 | 0.0508 | 0.9465 | 3.6 | 24.2 | 0.9754 | 2.83 | 0.9798 |
| Br-resin | 40 | 13.2 | 12.1 | 0.0687 | 0.9968 | 6.7 | 15.2 | 0.9938 | 1.85 | 0.9553 |
| | 80 | 24.3 | 20.6 | 0.0642 | 0.9817 | 4.6 | 27.0 | 0.9941 | 3.27 | 0.9448 |
| I-resin | 40 | 16.8 | 15.5 | 0.0878 | 0.9984 | 6.6 | 19.2 | 0.9983 | 2.40 | 0.9144 |
| | 80 | 36.6 | 33.6 | 0.0731 | 0.9951 | 2.5 | 42.4 | 0.9965 | 5.21 | 0.9386 |

### 3.3.3. Pseudo-Second Order Kinetics

The pseudo-second order kinetic model was also applied to the experimental data. It is given by the equation below [55]:

$$t/q_t = 1/k_2 q^2_e + (1/q_e) \times t \tag{6}$$

where $k_2$ (g mg$^{-1}$ min$^{-1}$) is the rate constant of a pseudo-second order adsorption reaction. Second order kinetics is said to be applicable if the plot of $t/q_t$ versus $t$ relationship shows linearity. Second order kinetics is more likely to predict the behavior over whole range of adsorption and is in agreement with chemical sorption being the rate-controlling step [62]. The graphical interpretation of the data by the pseudo-second order kinetic model is given in Figure 4 and the rate constants ($k_2$), correlation coefficients of the plots together with the experimental $q_e$ values are given in Table 2. The $q_e$ values obtained from the pseudo-second order model are also reasonably consistent with the experimental ones and the correlation coefficients are very high, indicating that chemisorption is also a determinant for the adsorption process [63].

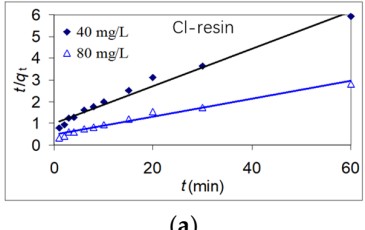
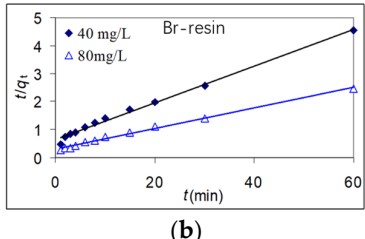
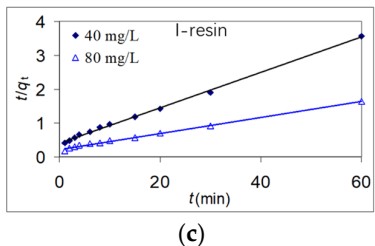

(**a**)                                      (**b**)                                      (**c**)

**Figure 4.** Pseudo-second order plots of $Cd^{2+}$ adsorption on $X^-$-loaded resins. (**a**) Cl-resin, (**b**) Br-resin, (**c**) I-resin.

### 3.3.4. Intra-Particle Diffusion Kinetics

The overall rate of ion exchange is often controlled by the intra-particle diffusion step with porous adsorbent and an intra-particle diffusion model is expressed in the equation given by Weber and Morris [64]:

$$q_t = k_{id} t^{1/2} \tag{7}$$

where $q_t$ is the quantity of metal ions adsorbed at time $t$ (mg g$^{-1}$) and $k_{id}$ is the intra-particle diffusion rate constant (mg g$^{-1}$ min$^{-1/2}$). Plots of $q_t$ versus $t^{1/2}$ are shown in Figure 5 and the $k_{id}$ and $R^2$ data are listed in Table 2. The correlation co-efficient data suggest that intra-particle diffusion plays a considerable part in the adsorption process but not the only rate controlling factor because the plots do not pass through the origin [65]. As pointed out above, ligand exchange may also play a significant role in determining the overall adsorption rate.

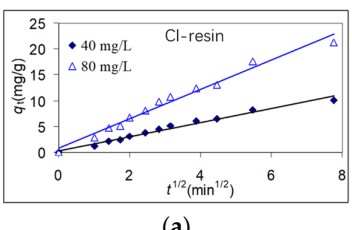
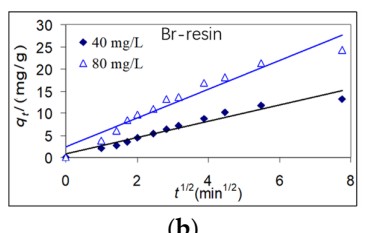
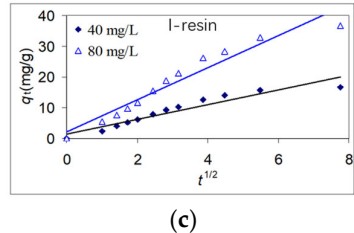

(**a**)                                      (**b**)                                      (**c**)

**Figure 5.** Intra-particle diffusion plots of $Cd^{2+}$ adsorption on $X^-$-loaded resins. (**a**) Cl-resin, (**b**) Br-resin, (**c**) I-resin.

### 3.4. Comparison of Dynamic Retention Efficiency of $Cd^{2+}$

The separation of $Cd^{2+}$ from other metal ions in this study is based on their different retention efficiency by the anion exchangers. The performance of the three modified resins were compared experimentally in the presence of some major metal ions including $Zn^{2+}$, $Cu^{2+}$ and $Pb^{2+}$ which are competing ions against $Cd^{2+}$ in the ion exchange process. Because ion exchange can be conveniently performed with a column, the comparative retention efficiency of $Cd^{2+}$ on $X^-$-loaded Amberlite IRA 900 resin or ZB-2 fibre in a dynamic process was studied by percolating a 20 mL feed solution containing 0.2% $Cd^{2+}$, 1% $Zn^{2+}$, 1% $Cu^{2+}$ and 0.1% $Pb^{2+}$ (*w/v*) at 1 mL min$^{-1}$ through an ion exchange column packed with 0.5 g of dry resin or fibre, which gives a wet volume of 1 mL after saturation with water. $Cd^{2+}$ in the effluent was analyzed and its retention efficiency was calculated by mass balance and the results are given in Table 3. The retention of $Cd^{2+}$ increased in the order of Cl-resin < Br-resin < I-resin. The fibres have much higher retention efficiency than the corresponding resins under the same operational conditions. This could be attributed to their different properties. The fibre has much higher adsorption capacity than the resin, i.e., 3 meq g$^{-1}$ versus 1 meq g$^{-1}$. It also has the functional groups on the surface, thus intra-particle diffusion was not involved in the adsorption process.

Although the retention efficiency obtained in most of the experiments is not sufficient for complete separation of $Cd^{2+}$ from other metal ions under the operational conditions,

this can be compensated by using different strategies. Potential ways of improving the efficiency include using larger column volume, optimization of the feeding flow rate, re-percolating the effluent through the same column before washing and elution, and use of multi-stage adsorption system. The data in the third row of Table 3 show that after re-percolating the effluent, the total retention efficiency improved considerably. We can assume from these results that quantitative $Cd^{2+}$ retention could be attained by using a lower flow rate or multi-stage adsorption. In addition, when $Cd^{2+}$ concentration in the feed solution was 200 mg $L^{-1}$, retention efficiency was all over 99%, implying that efficient adsorption can also be achieved when the initial $Cd^{2+}$ concentration is low.

It is worth mentioning that $Zn^{2+}$ could have adverse impact on the retention of $Cd^{2+}$ and this should be especially profound for feed solutions with high $Zn^{2+}$ concentration and when using $Cl^-$- and $Br^-$-loaded exchangers due to the competition of $Zn^{2+}$ against $Cd^{2+}$ for ion exchange sites. Moreover, $Cu^{2+}$ has significant impact on $Cd^{2+}$ adsorption by $I^-$-loaded exchangers as further discussed in the following section. Therefore, the retention efficiency of $Cd^{2+}$ for real samples should be examined according to the concentrations of these co-existed elements.

**Table 3.** Retention efficiency of $Cd^{2+}$ by $X^-$-loaded resin and fibre.

| Ion Exchanger | Cl-Resin | Br-Resin | I-Resin | Cl-Fibre | Br-Fibre | I-Fibre |
|---|---|---|---|---|---|---|
| Retention efficiency (%) | 66.3 | 67.1 | 94.5 | 81.6 | 86.8 | 99.5 |
| Retention efficiency after re-percolating the effluent (%) | 84.3 | 89.3 | 99.1 | | | |

### 3.5. Effect of Acidity

The effect of acidity on the $Cd^{2+}$ adsorption was studied by using different HCl concentrations in the feed solution, and the results are presented in Figure 6. It is evident that the HCl concentration had significant effect on the $Cd^{2+}$ adsorption efficiency. We can assume that the $Cl^-$ concentration is the major factor determining the $Cd^{2+}$ adsorption. It is clear that when the HCl concentration was under 0.5 M, the adsorption had constant efficiency. With the increase in HCl concentration, the adsorption became less efficient. In 6 M HCl, the adsorption was very low. This can be attributed to the equilibrium between the solid resin phase and the liquid feed solution. In addition, high HCl concentration (>1 M) leads to oxidation of $I^-$ by air, but 0.1% ascorbic acid dissolved in the feed solution effectively eliminated the oxidation. Although the HCl concentration has a profound influence on the adsorption, this method tolerated much higher acidity than solvent extraction and cementation.

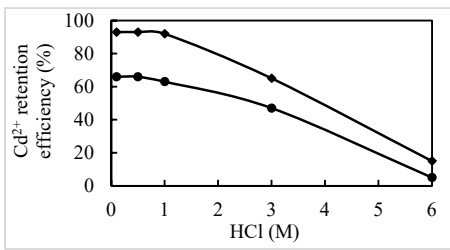

**Figure 6.** Effect of HCl concentration on $Cd^{2+}$ adsorption efficiency on Br-resin (●) and I-resin (♦).

### 3.6. Removal of Retained Foreign Metal Ions

To separate the co-existed elements from $Cd^{2+}$ as completely as possible, chemically and physically retained ions of these elements are to be removed prior to the elution of $Cd^{2+}$. $Mn^{2+}$, $Fe^{3+}$, $Co^{2+}$, $Ni^{2+}$, $Cu^{2+}$, $Zn^{2+}$, and $Pb^{2+}$ were examined in this study as they are the most commonly encountered elements with Cd in minerals and mineral processing residues. Some of these elements form halo-complexes with appreciable stability and could

be chemically retained by the exchangers. At the same time, the residual feed solution left in the pores and voids of the resin phase also contains considerable quantities of these elements. To remove them prior to the $Cd^{2+}$ stripping stage, a washing step was applied following the sample loading by using 0.10 M HCl. It was found that $Mn^{2+}$, $Fe^{3+}$, $Co^{2+}$, and $Ni^{2+}$ were readily removed by washing with 20 mL of 0.1 M HCl from the 1 mL resin or fiber column. $Cu^{2+}$ and $Pb^{2+}$ were also removed easily from $Cl^-$- or $Br^-$-loaded resins. This is due to the weak complexation of these ions with the ligands. However, $Cu^{2+}$ can be easily reduced to $Cu^+$ by $I^-$, resulting in consumption of the latter and strong retention of $Cu^+$ via formation of stable complex with $I^-$ and/or CuI precipitate. In addition, the transformation of $I^-$ to $I_2$ could deteriorate the performance of $I^-$-loaded resin and fibre. Therefore, the separation of $Cu^{2+}$ by $I^-$-loaded resin or fibre was not considered. Coordination between $I^-$ and $Pb^{2+}$ has considerable strength [52] and $Pb^{2+}$ was partially retained by $I^-$-loaded resin or fibre and resulted in incomplete separation from $Cd^{2+}$. Stability of $Zn^{2+}$ complexes decreases in the order of $[ZnCl_4]^{2-} > [ZnBr_4]^{2-} > [ZnI_4]^{2-}$ [50] and its retention efficiency declined in the same order. Compared with the other co-existed elements, halo-complexes of $Zn^{2+}$ have higher stability and competitiveness against $Cd^{2+}$ in the ion exchange process. Nevertheless, experimental results have shown that 40 mL of the washing solution completely removed $Zn^{2+}$. Under these conditions, $Cd^{2+}$ was not lost from the exchangers. The use of HCl solution higher than 0.1 M was less effective in the removal of $Zn^{2+}$ whereas a HCl lower than 0.1 M caused the loss of $Cd^{2+}$ in the washing stage.

### 3.7. Stripping of $Cd^{2+}$

A common strategy for stripping $Cd^{2+}$ from anion exchanger is using highly concentrated anions such as $NO_3^-$ ($HNO_3$ or $NaNO_3$), but consumption of such eluent is massive due to the strong retention of $Cd^{2+}$ on the anion exchanger [38,45,50,51,61]. In this study, we used water to disequilibrate the formation of $Cd^{2+}$ halo-complex on the exchanger by diluting the $Cl^-$ concentration in the system. This showed that 50–68% of the adsorbed $Cd^{2+}$ was eluted with 20 bed-volumes (20 mL) of water (Figure S2). The stripping efficiency was defined as the stripped fraction as a percentage of the total retained amount. More water did not increase the stripping efficiency considerably. In general, the stripping efficiency with water was in reverse order with the stability of the halo-complexes. Although the stripping is not sufficient under the experimental conditions, this technique enjoys the benefit of environmental-benign asset.

Another stripping strategy is to use a ligand which forms positively charged species with $Cd^{2+}$ so that its halo-complex can be decomposed. Ammonia is a typical candidate for this purpose [38,45] and has the potential to accelerate the stripping compared with water by forming $[Cd(NH_3)_4]^{2+}$ which spontaneously dissociates from the $C-N(CH_3)^+$ groups of the exchanger. As shown in Figure S3, 10 bed-volumes (10 mL) of 0.5 M $NH_3$ can strip 50% to 60% of the adsorbed $Cd^{2+}$, much more efficient than water. However, a further increase in the volume of ammonia solution did not make appreciable improvement in the stripping efficiency. This could be due to the formation of $Cd(OH)_2$ precipitate on the surface of the exchanger which hindered further stripping of $Cd^{2+}$. It also shows that the different stability of the halo-complexes and exchangers (i.e., resin vs. fibre) did not cause significant difference in stripping efficiency and this could be attributed to the high stability of $[Cd(NH_3)_4]^{2+}$ [52].

Compared to ammonia, amines should have higher stripping ability due to the formation of more stable complex with $Cd^{2+}$ [52] and this should be more promising with multidentate ligands such as ethylenediamine. Stripping results with En are shown in Figure S4. Compared with $H_2O$ and $NH_3$, En is a much more efficient eluent because only 5 bed-volumes (5 mL) of 0.3 M En were consumed to strip over 90% of $Cd^{2+}$ from $I^-$-loaded fibre. This is attributed to the formation of highly stable chelate $[CdEn_2]^{2+}$ and its spontaneous dissociation from the exchanger.

### 3.8. Separation Performance

The separation performance was investigated by percolating 20 mL of multielement feed solution at 1 mL min$^{-1}$ through a column of 1 mL wet volume filled with halide loaded Amberlite IRA 900 resin or ZB-2 fibre followed by washing out the residual co-existed elements with 20 mL of 0.10 M HCl and eluting the adsorbed Cd$^{2+}$. Elements in the effluent, wash-out, and eluate were determined by ICP-MS and the recovery of the elements were calculated by taking the percentage of the recovered amount in the stripped solution over the total input amount in the feed solution. The concentrations of the elements in the feed solution were 0.1% Cd, 1% Zn, 1% Cu, 1% Pb, 0.1% Mn, 0.5% Fe, 0.1% Co, and 0.1% Ni (*w/v*), respectively. The results are given in Tables 4 and 5. It is clear that Zn$^{2+}$ is the most difficult element for separation from Cd$^{2+}$, followed by Cu$^{2+}$ and Pb$^{2+}$. In comparison, Mn$^{2+}$, Fe$^{3+}$, Co$^{2+}$ and Ni$^{2+}$ were easily separated from Cd$^{2+}$ due to their low halo-complex stability. The separation of Cu$^{2+}$ by I$^{-}$-loaded resin was not examined due to its reduction by I$^{-}$.

Considering the difficulty in complete Zn$^{2+}$ separation from Cd$^{2+}$, a separate experiment was conducted which involved the use of a feed solution containing 0.1% Cd$^{2+}$ and 1% Zn$^{2+}$, 1 mL of wet I$^{-}$-loaded anion exchange fibre, 40 mL of 0.1 M HCl as the washing agent and En solutions as the stripping agents. The results given in Table 6 show that by using these parameters, Zn$^{2+}$ was separated from Cd$^{2+}$ with high efficiency.

**Table 4.** Metal recovery by water elution.

| Element | Element Recovery in Eluate (% of Total Input) | | | | | |
|---|---|---|---|---|---|---|
| | **Cl-Resin** | **Br-Resin** | **I-Resin** | **Cl-Fibre** | **Br-Fibre** | **I-Fibre** |
| Cd$^{2+}$ | 24.18 | 24.39 | 24.20 | 63.86 | 61.63 | 49.28 |
| Zn$^{2+}$ | 1.85 | 1.67 | 0.98 | 0.24 | 0.08 | 0.06 |
| Cu$^{2+}$ | ND [a] | ND | NE [b] | ND | 0.02 | NE |
| Pb$^{2+}$ | ND | ND | 2.5 | ND | ND | 7.5 |
| Mn$^{2+}$ | ND | ND | ND | ND | ND | ND |
| Fe$^{3+}$ | ND | ND | ND | ND | ND | ND |
| Co$^{2+}$ | ND | ND | ND | ND | ND | ND |
| Ni$^{2+}$ | ND | ND | ND | ND | ND | ND |

Colum size 1 mL, 40 mL H$_2$O for elution, [a] Not detectable, [b] Not examined.

**Table 5.** Metal recovery by NH$_3$ elution.

| Element | Element Recovery in Eluate (% of Total Input) | | | | | |
|---|---|---|---|---|---|---|
| | **Cl-Resin** | **Br-Resin** | **I-Resin** | **Cl-Fibre** | **Br-Fibre** | **I-Fibre** |
| Cd$^{2+}$ | 35.45 | 37.30 | 67.15 | 60.45 | 54.40 | 68.20 |
| Zn$^{2+}$ | 2.12 | 1.79 | 1.19 | 0.30 | 0.24 | 0.20 |
| Cu$^{2+}$ | 0.023 | 0.029 | NE | 0.036 | 0.066 | NE |
| Pb$^{2+}$ | 0.06 | 0.04 | 0.13 | <0.01 | <0.01 | 0.17 |
| Mn$^{2+}$ | ND | ND | ND | ND | ND | ND |
| Fe$^{3+}$ | ND | ND | ND | ND | ND | ND |
| Co$^{2+}$ | ND | ND | ND | ND | ND | ND |
| Ni$^{2+}$ | ND | ND | ND | ND | ND | ND |

* 1 mL wet column volume, 10 mL 0.5 M NH$_3$ for elution.** Cu$^{2+}$ was not examined by I$^{-}$-loaded columns due to its reduction.

**Table 6.** Cd and Zn recovery (% of total input) by En elution from I$^{-}$-loaded ZB-2 fibre.

| En Concentration (M) | Cd$^{2+}$ Recovery (%) | Zn$^{2+}$ Recovery (%) | Separation Factor ($f_{Cd/Zn}$) |
|---|---|---|---|
| 0.1 | 71.73 | 0.13 | 5518 |
| 0.2 | 88.45 | 0.04 | 22,113 |
| 0.3 | 97.03 | 0.02 | 48,515 |

* 1 mL column volume and 6 mL En for elution.

It is worth-noting that higher concentration of En led to higher $Cd^{2+}$ recovery from the $I^-$-loaded fibre but lower $Zn^{2+}$ recovery although the $[ZnEn_2]^{2+}$ complex has higher stability than $[CdEn_2]^{2+}$ ($\log K_f$ 14.11 for $[ZnEn_2]^{2+}$ and 12.09 for $[CdEn_2]^{2+}$, respectively) [52]. This was possibly caused by the much higher stability of Zn-OH complex than that of Cd-OH complex ($\log K_f$ 17.66 for Zn-OH complex and 8.62 for Cd-OH complex, respectively) [52] which led to the relatively stable retention of $Zn^{2+}$ by ion exchange or precipitation due to the basic property of the En solution and lower stripping efficiency. This is desirable for complete separation of the two elements.

*3.9. Comparison with Conventional and Reported Techniques*

Compared with some typical hydrometallurgical techniques applied in industry and reported in the literature, several advantages can be recognized in the present method. Solvent extraction and cementation are the two typical techniques for Cd separation applied in industry. In comparison with solvent extraction, the present technique eliminated the use of organic solvents, and it's also much more tolerant to high acidity. The benefit of the latter feature underlies its value for industrial application because mineral raw materials are often leached with acid. Compared with cementation, the present technique is also much less affected by high acidity, and there is no zinc dust consumption and Cd is separated from the matrix as a pure element in this method, thus no slag is produced in its removal process and no further separation from other metals is needed for its recovery. Many other techniques, such as imprinted sorbent separation, polymer inclusion membrane extraction, and water-based two-phase system extraction, have been only practiced in lab-scale study due to their limitations, whereas the present method is highly scalable for industrial application given the availability of commercial ion exchanger products.

**4. Conclusions**

By pre-loading strongly basic anion exchange resin and fibre with halide ($Br^-$ and $I^-$), higher retention efficiency for $Cd^{2+}$ from $Cl^-$ solution was achieved by forming mixed ligand complexes than using Cl-formed exchanger. Thus, no excess $Br^-$ or $I^-$ is needed in the feed solution, and the operating cost can be minimized. Adsorption isotherms showed that the adsorption capacity increases in the order of Cl-resin < Br-resin < I-resin. The Langmuir model fitted well for Cl-resin and Br-resin while the Freundlich model fitted the I-resin better than Langmuir model. $[CdX_2Cl_2]^{2-}$ (X=Cl, Br or I) was assumed to be the dominant species retained by the exchangers, but different species could be formed on $I^-$-loaded resin and fibre. Kinetics data of all the resins were well described by both Langergren pseudo-first order model and pseudo-second order model, implying that both chemical and physical factors are important for the sorption. Intra-particle diffusion also plays a significant role in the adsorption process with the resin. Under the optimized conditions, $Mn^{2+}$, $Fe^{3+}$, $Co^{2+}$, and $Ni^{2+}$ were not retained by any of the ion exchangers and thus separated from $Cd^{2+}$ completely. $I^-$-loaded exchangers exhibited superior performance for the separation of $Cd^{2+}$ from $Zn^{2+}$ over $Cl^-$- or $Br^-$-loaded exchangers, whereas the latter ones are more ideal for separating $Cd^{2+}$ from $Cu^{2+}$ and $Pb^{2+}$. Water exhibited appreciable stripping capability for $Cd^{2+}$ ions from the resin or fibre and ammonia accelerated the elution considerably, but the overall stripping efficiency of $Cd^{2+}$ by water or ammonia was not sufficient. Ethylenediamine showed much higher stripping capability with elution efficiency of over 90% under adequate conditions. This study represents a potentially scalable, environmentally friendly, and cost-effective technology for the separation of cadmium from base metals compared with conventional techniques.

**Supplementary Materials:** The following supporting information can be downloaded at: https://www.mdpi.com/article/10.3390/pr11041051/s1, Figure S1: IR spectra of halide and $[CdX_4]$ loaded anion exchange resins; Figure S2. Stripping efficiency of $Cd^{2+}$ with 20 bed volume of $H_2O$ from $X^-$-loaded Amberlite IRA 900 resin; Figure S3. Stripping efficiency of $Cd^{2+}$ with 10 bed volume of

0.5 M $NH_3$ from $X^-$ loaded Amberlite IRA 900 resins; Figure S4. Stripping efficiency of $Cd^{2+}$ with 5 bed volume of ethylenediamine from $I^-$-loaded ZB-2 fibre.

**Author Contributions:** Conceptualization: Y.Z., C.H., X.D. and Y.W.; methodology: Y.Z., C.H. and Y.W.; resources: C.H. and Y.W.; Investigation: Y.Z., X.D. and G.D.; data processing: Y.Z. and G.D.; writing, manuscript drat preparation: Y.Z.; writing—review and editing: Y.W. and C.H.; funding acquisition: Y.W. All authors have read and agreed to the published version of the manuscript.

**Funding:** This research was funded by National Natural Science Foundation of China (No. 22178350).

**Data Availability Statement:** The research created experimental data that can be found in the tables and figures presented in this manuscript.

**Acknowledgments:** The authors would like to acknowledge the funding provided by National Natural Science Foundation of China (No. 22178350).

**Conflicts of Interest:** The authors declare no conflict of interest.

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
