# Peer review of "Green and Cost-Effective Separation of Cadmium from Base Metals in Chloride Medium with Halide-Loaded Anion Exchanger"

_processes, doi:10.3390/pr11041051_

Round 1
Reviewer 1 Report
Dear Authors
The proposed work is interesting and is the requirement of chemical processes for development of robust and accurate analytical method for the separation of Cadmium metal from coexisting alike metals. The work needs some substantial improvements before publication in the Journal Processes. Therefore, I recommend major revision of the current form of manuscript prior publication
The work shows no characterization of the anion exchangers, no surface areas, pHzc. The work lacks identification of exact adsorption mechanism through the assistance of a suitable technique such as FTIR or titrimetric analysis.
Page 4, line 193 why Zn, Cu were taken as 0.1%, Fe as 0.5% and other metals as 0.1% were taken for separation performance experiments. The logic of selecting different compositions of these metals is not clear.
Equation 4 must be corrected by writing ln (natural log) instead of lg.
Page 6,lines 247-250, how authors interpret formation of [CdBr2(Cl2)], [CdI2(Cl)2] and [CdI(Cl)3] complexes on adsorption on halide loaded anion exchangers. Any characterization tool?
In kinetics study why two different metal ion concentrations were selected. Apparently it seems that higher concentration of Cd2+ are favorable for kinetics experiment. But the results were not properly interpreted. What was the effect of higher concentration of kinetic parameters and what information do these higher values furnish.
Page 13, line 361, explanation of greater retention efficiency of fibers is insufficient. What does this mean “kinetic advantage over the resins”. Is this due to greater surface area factor? Or something else?
Page 15, what was the volume of 0.5 M ammonia solution equivalent to 10 bed-volumes? Similarly, what was the volume of 0.3 M ethylene diamine equivalent to 5 bed-volumes.
Table 6, molecular formula of ethylene diamine is incorrect. Please make correction. The table also looks incompletely made. Add a row and merge to label the different concentrations of ethylene diamine.
Compare the current method with previously reported extraction techniques such as SPE, SPME, cloud point extractions, solidification of floating drop method and other modern techniques. What is the sensitivity and precision of the current method.
Author Response
Dear Authors
The proposed work is interesting and is the requirement of chemical processes for development of robust and accurate analytical method for the separation of Cadmium metal from coexisting alike metals. The work needs some substantial improvements before publication in the Journal Processes. Therefore, I recommend major revision of the current form of manuscript prior publication
The work shows no characterization of the anion exchangers, no surface areas, pHzc. The work lacks identification of exact adsorption mechanism through the assistance of a suitable technique such as FTIR or titrimetric analysis.
Response: The anion exchangers are commercial products that have the specifications given by the manufacturers and hence, it may be not necessary to re-characterize their physical properties. However, some of the important properties, including the particle size, pore size and exchange capacity (the resin), the linear density and exchange capacity (the fiber) have been given in the paper.
To elucidate the adsorption mechanism, FTIR has been used to characterize the resin after loading with -OH, -I, and [CdX4] complex of prepared under different conditions.
Page 4, line 193 why Zn, Cu were taken as 0.1%, Fe as 0.5% and other metals as 0.1% were taken for separation performance experiments. The logic of selecting different compositions of these metals is not clear.
Response: Considering this is a fundamental study, the authors think the use of such concentrations is representative and reasonable for average situation.
Equation 4 must be corrected by writing ln (natural log) instead of lg.
Response: has been corrected as suggested.
Page 6,lines 247-250, how authors interpret formation of [CdBr2(Cl2)], [CdI2(Cl)2] and [CdI(Cl)3] complexes on adsorption on halide loaded anion exchangers. Any characterization tool?
Response: FTIR has been used to characterize the resins after loading Cd under different conditions which should correspond to formation of different complex species, and relevant discussion has been made.
In kinetics study why two different metal ion concentrations were selected. Apparently it seems that higher concentration of Cd2+ are favorable for kinetics experiment. But the results were not properly interpreted. What was the effect of higher concentration of kinetic parameters and what information do these higher values furnish.
Response: In adsorption kinetics study, multiple concentrations of the adsorbate are often used for model validation (Hamidpour, M. et. al. Rev. Int. Contam. Ambie. 34 (2) 307-316, 2018, DOI: 10.20937/RICA.2018.34.02.11; P. Senthil Kumar et. al. Brazil. J. Chem. Eng., 27(2), 347-355, 2010; Jongho Kim et. al. Polymers, 2019, 11, 297, doi:10.3390/polym11020297). In this work, the experimental results of the two initial Cd concentrations have been interpreted in the text.
Page 13, line 361, explanation of greater retention efficiency of fibers is insufficient. What does this mean “kinetic advantage over the resins”. Is this due to greater surface area factor? Or something else?
Response: According to the manufacturer’s specification, the ZB-2 anion exchange fibre used in this study has the functional groups on its outer surface and thus, intraparticle diffusion is not involved in ion exchange process. In comparison, the anion exchange resin used in this study has porous structure and henc, the ion exchange process involves intra-particle diffusion. As a consequence, it is reasonable to accept that the fiber has kinetic advantage over the resin, as has been verified in the study. Thie view has been added to the revised version.
Page 15, what was the volume of 0.5 M ammonia solution equivalent to 10 bed-volumes? Similarly, what was the volume of 0.3 M ethylene diamine equivalent to 5 bed-volumes.
Response: The bed volume and the corresponding volumes of the stripping agents have been clarified in the revised version.
Table 6, molecular formula of ethylene diamine is incorrect. Please make correction. The table also looks incompletely made. Add a row and merge to label the different concentrations of ethylene diamine.
Response: corrections have been made for the molecular formula. Table 6 has been modified for clarity as suggested by the reviewer.
Compare the current method with previously reported extraction techniques such as SPE, SPME, cloud point extractions, solidification of floating drop method and other modern techniques. What is the sensitivity and precision of the current method.
Response: The previously reported techniques which can be found in the literature have been described and discussed in the Introduction, and the current method’s advantages have been pointed out. This technique aims at separating Cd from base metals for industrial purpose, rather than for analysis. So, sensitivity and precision are irrelevant.
Reviewer 2 Report
In this manuscript, an environmentally friendly and economic technology for Cd2+ separation from base metals in chloride medium with halide-loaded anion exchanger was proposed and adsorption thermodynamic and kinetics were also investigated in detail. Besides, the stripping of Cd2+ in the complex solution with different agents was explored in this work. However, some aspects should still be taken into account. Details are listed as follows:
1. In Introduction, the author wrote "Compared with pyrometallurgical processes, hydrometallurgical technologies are more appropriate from environmental and economic perspectives in the separation of cadmium because less energy is consumed and less dust collecting and gas cleaning are needed". However, there are few discussions about the pyrometallurgical processes mentioned, and the advantages and disadvantages of this work are not highlighted. Thus, I suggested that the author added the related technologies and data to support this description.
2. Besides, the author should exhibit more information about the origin of the feed solution (0.1% Cd, 1% Zn, 1% Cu, 0.1% Pb, 0.1% Mn, 0.5% Fe, 0.1% Co, 0.1% Ni (w/v), and 0.15 M HCl) used in the experiments and supplement more details.
3. In stripping of Cd2+, three kinds of stripping agents (water, ammonia, ethylenediamine) were investigated. Among them, the stripping efficiencies of Cd2+ were lower than 70% using water and ammonia as stripping agents, while there is a high recovery efficiency of Cd2+, approximately 90%. Could be of interest to know the advantages and mechanism of ethylenediamine in comparison to water and ammonia? And the corresponding figures and tables should be shown in the manuscript.
Author Response
In this manuscript, an environmentally friendly and economic technology for Cd2+ separation from base metals in chloride medium with halide-loaded anion exchanger was proposed and adsorption thermodynamic and kinetics were also investigated in detail. Besides, the stripping of Cd2+ in the complex solution with different agents was explored in this work. However, some aspects should still be taken into account. Details are listed as follows:
- In Introduction, the author wrote "Compared with pyrometallurgical processes, hydrometallurgical technologies are more appropriate from environmental and economic perspectives in the separation of cadmium because less energy is consumed and less dust collecting and gas cleaning are needed". However, there are few discussions about the pyrometallurgical processes mentioned, and the advantages and disadvantages of this work are not highlighted. Thus, I suggested that the author added the related technologies and data to support this description.
Response: As suggested by the reviewer, a more detailed discussion has been added on pyrometallurgical techniques for Cd separation with particular processes and data involved in the Introduction.
- Besides, the author should exhibit more information about the origin of the feed solution (0.1% Cd, 1% Zn, 1% Cu, 0.1% Pb, 0.1% Mn, 0.5% Fe, 0.1% Co, 0.1% Ni (w/v), and 0.15 M HCl) used in the experiments and supplement more details.
Response: This is just a fundamental study on the separation technique and the objective is to develop a feasible technique for Cd separation from multiple metal materials. The multi-element feed solution was a synthetic sample and the concentrations were chosen based on the average situations encountered in hydrometallurgical industry, rather than from any special raw material. This has been further clarified in the revised version.
- Instripping of Cd2+, three kinds of stripping agents (water, ammonia, ethylenediamine) were investigated. Among them, the stripping efficiencies of Cd2+ were lower than 70% using water and ammonia as stripping agents, while there is a high recovery efficiency of Cd2+, approximately 90%. Could be of interest to know the advantages and mechanism of ethylenediamine in comparison to water and ammonia? And the corresponding figures and tables should be shown in the manuscript.
Response: The use of water and ammonia for stripping was mainly for comparison. The insufficient stripping with them can be improved by using more efficient parameters as has been discussed in the text. The high Cd recovery was achieved with the fiber. In comparison, by using the resin, lower Cd recovery was achieved. This is a good indication to the advantages of the fiber over the resin
As has been discussed in the text, as a multidentate ligand, ethylenediamine can for stable chelate with Cd, which underlines the advantages of it over H2O and NH3. In comparison, H2O does not form stable species with Cd, but dilute the Cl concentration in the system and disequilibrate the [CdX4]2- formation reaction. NH3 for stable positively charged complex [Cd(NH3)4]2+ with Cd and facilitates the dissociation of Cd from the positively charged functional groups of the exchanger.
Reviewer 3 Report
Referee report on manuscript “Green and Cost-effective Separation of Cadmium from Base Metals in Chloride Medium with Halide-Loaded Anion Exchanger” by Yanlin Zhang et al
This version does not look worthy and cannot be recommended for publication in this form and at least needs some proper improvement and clarification.
1. Introduction. The relevance and novelty of the work is not disclosed. It is not at all clear how much this topic is still interesting and what has been done in this direction in recent years. Only two links on [25] of 2014 and [26] of 2022 are under 10 years old.
2. Line 101-105. Note about [CdI4]2−. It is important to remark here that the formation of (CdIi)n clusters have been studied in details by Bellucci et al, using SEM, optical absorption, luminescence and corresponding computer modelling, in:
S Bellucci et al 2007 J. Phys.: Condens. Matter 19 395015
DOI 10.1088/0953-8984/19/39/395015
3. Line 51. The end of the sentence need supporting references for each metals. In the case of Bi-metal, its behaviors in CdI2 was recently detailed by Karbovnyk, I., et al. "BiI3 nanoclusters in melt-grown CdI2 crystals studied by optical absorption spectroscopy." Physica B: Condensed Matter 413 (2013): 12-14, while Pb –ions in
Novosad, S. S., et al. "Spectral and kinetic characteristics of CdI2 and CdI2: Pb scintillators." Journal of Applied Spectroscopy 75 (2008): 826-831.
4. Experimental. Why optical, luminescence and infrared measurements were not carried out?
5. Figure 1. For greater clarity, information about Cl-resin; Br-resin and I-resin would be useful to place respectively on each of the drawings (a, b, or c).
6. The same for Figures 2, 3, 4 and 5.
7. Tables 4, 5 and 6. Is it possible to specify the charge states of metals given in these Tables?
Author Response
This version does not look worthy and cannot be recommended for publication in this form and at least needs some proper improvement and clarification.
- The relevance and novelty of the work is not disclosed. It is not at all clear how much this topic is still interesting and what has been done in this direction in recent years. Only two links on [25] of 2014 and [26] of 2022 are under 10 years old.
Response: The novelty of the technique has been further emphasized in the revised version, and the literature has been updated as suggested by the reviewer.
- Line 101-105. Note about [CdI4]2−. It is important to remark here that the formation of (CdIi)n clusters have been studied in details by Bellucci et al, using SEM, optical absorption, luminescence and corresponding computer modelling, in:
S Bellucci et al 2007 J. Phys.: Condens. Matter 19 395015
DOI 10.1088/0953-8984/19/39/395015
Response: This study is about the selective separation of Cd from other base metals for clean production and environment purposes. The study involves complex formation but is not dealing with crystal structure, cluster formation, or optical properties. The authors suppose that there are no clusters formed. Complex species have been characterized with FTIR as suggested by other reviewers and added to the revised version.
- Line 51. The end of the sentence need supporting references for each metals. In the case of Bi-metal, its behaviors in CdI2 was recently detailed by Karbovnyk, I., et al. "BiI3 nanoclusters in melt-grown CdI2 crystals studied by optical absorption spectroscopy." Physica B: Condensed Matter413 (2013): 12-14, while Pb –ions in
Novosad, S. S., et al. "Spectral and kinetic characteristics of CdI2 and CdI2: Pb scintillators." Journal of Applied Spectroscopy 75 (2008): 826-831.
Response: Again, this study deals with selective separation of Cd from other base metals (excluding Bi) for metallurgical industry and/or environment protection and aims at processing general industrial raw materials and wastes. Supporting references have been cited for general cases as suggested by the reviewer but not for each-co-existed element.
The literature given by the reviewer are in different scientific area and irrelevant to this study and therefore not considered.
- Why optical, luminescence and infrared measurements were not carried out?
Response: As has been clarified, this study does not deal with crystal structure and optical properties. As such, such optical characterization was not carried out. But FTIR characterization was performed to elucidate the complex species adsorbed on the anion exchanger.
- Figure 1. For greater clarity, information about Cl-resin; Br-resin and I-resin would be useful to place respectively on each of the drawings (a, b, or c).
Response: Legend has bee added as suggested.
- The same for Figures 2, 3, 4 and 5.
Response: legend has been added.
- Tables 4, 5 and 6. Is it possible to specify the charge states of metals given in these Tables?
Response: oxidation states have been added as suggested by the reviewer.
Round 2
Reviewer 1 Report
The authors have not made sufficient revision and have ignored many suggestions which this reviewer thinks poor response from author side.
No peak labeling in FTIR? No solid evidence of formation of [CdBr2(Cl2)], [CdI2(Cl)2] and [CdI(Cl)3].
No comparison with other reported methods and no any clear justification for not following the suggestion. A research paper should highlight importance of your work how and why your work should be given preference. You have not put light on the specialty in your method.
Authors also denied to find precision and sensitivity of the method by arguing that the work is not analytical. It should be mentioned that the separation work is analytical and it is necessary to know how lowest possible concentration of Cd could be separated from multi metal system.
What is the influence of pH on separation?
It is again urged to revise the manuscript properly according to previous and current suggestions
Author Response
Reviewer 1
|
Yes |
Can be improved |
Must be improved |
Not applicable |
|
|
Does the introduction provide sufficient background and include all relevant references? |
( ) |
(x) |
( ) |
( ) |
|
Are all the cited references relevant to the research? |
( ) |
(x) |
( ) |
( ) |
|
Is the research design appropriate? |
( ) |
(x) |
( ) |
( ) |
|
Are the methods adequately described? |
( ) |
(x) |
( ) |
( ) |
|
Are the results clearly presented? |
( ) |
(x) |
( ) |
( ) |
|
Are the conclusions supported by the results? |
( ) |
(x) |
( ) |
( ) |
Comments and Suggestions for Authors
The authors have not made sufficient revision and have ignored many suggestions which this reviewer thinks poor response from author side.
Response: As suggested by the reviewer, we have enhanced the Introduction by covering more background information to support the rationality of our methodology. We have deleted a few irrelevant references. And we have modified the description of the Experimental and Results and Discussion to make the paper more informative and readable.
No peak labeling in FTIR? No solid evidence of formation of [CdBr2(Cl2)], [CdI2(Cl)2] and [CdI(Cl)3].
Response: The key IR peaks have been labelled as suggested.
We agree that the IR spectra cannot provide solid evidence of the complex composition. So, we have done elemental analysis to quantify the ratio of Br/Cd and I/Cd in the complexes adsorbed on the resins. From the molar ratio results presented and discussed in the text (Section 3.2), we think the assumed complex composition can be justified by the experimental results.
No comparison with other reported methods and no any clear justification for not following the suggestion. A research paper should highlight importance of your work how and why your work should be given preference. You have not put light on the specialty in your method.
Response: As suggested by the reviewer, we have added a separate paragraph at the end of Results and Discussion (Section 3.9) to compare our technique with some typically applied and reported techniques to justify the advantages of our technique. We have also highlighted the importance and specialty of our method in the Introduction and Conclusions in the revised version.
Authors also denied to find precision and sensitivity of the method by arguing that the work is not analytical. It should be mentioned that the separation work is analytical and it is necessary to know how lowest possible concentration of Cd could be separated from multi metal system.
Response: One of the authors (Yanlin Zhang) had a paper published using the same method principle for online separation of Cd from base metals and its quantitative analysis by vapor generation-atomic absorption spectrometry in which the sensitivity and precision have been given and the data have been cited in the newly revised version (Ref. 53: Samuel B. Adeloju, Yanlin Zhang, Anal. Chem., 2009, 81, 4249-4255. https://doi.org/10.1021/ac802618u). The present paper is to further clarify some fundamental and mechanistic issues of the adsorption process for industrial application.
What is the influence of pH on separation?
Response: The effect of acidity has been studied and the results are presented in Section 3.5.
It is again urged to revise the manuscript properly according to previous and current suggestions.
Response: We have added a new paragraph (the underlined paragraph in the Introduction) and some new sentences in the last paragraph of the Introduction to make the paper more cohesive.

Reviewer 3 Report
The author have reasonably improved their original manuscript.
Paper can be accepted.
Author Response
The reviewer has agreed to accept the manuscript.
Round 3
Reviewer 1 Report
I recommend the publication of revised version.